# Methyl Jasmonate Enhances the Resistance of *Populus alba* var. *pyramidalis* Against *Anoplophora glabripennis* (Coleoptera: Cerambycidae)

**DOI:** 10.3390/insects16020153

**Published:** 2025-02-03

**Authors:** Pengpeng Shao, Jiayu Luo, Rui Zhang, Jianfeng Liu, Dandan Cao, Zhi Su, Jianrong Wei

**Affiliations:** 1School of Life Sciences, Hebei University, Baoding 071002, China; shaopp1225@163.com (P.S.); m19331284156@163.com (J.L.); zzhang2022132@163.com (R.Z.); jianfengliu@hbu.edu.cn (J.L.); 2Research Center of Biotechnology, Hebei University, Baoding 071002, China; caodandan666@163.com; 3Experimental Center of Desert Forest, Chinese Academy of Forestry, Denkou, Bayannaoer 015200, China; slzxsuzhi@163.com

**Keywords:** *Anoplophora glabripennis*, poplar, methyl jasmonate, induced defense, wood borer

## Abstract

The Asian longhorned beetle *Anoplophora glabripennis* (ALB) is a serious wood borer of hard-wood trees. *Populus alba* var. *pyramidalis* (PaP) is the predominant shelterbelt tree species in northwestern areas of China and has been threatened by ALB. A feasible way to protect PaP from ALB is to improve its ability to resist pest attacks. Though the effect of MeJA has been tested in many plants, this is the first case thoroughly studied in a poplar species against a wood borer. Because it is a challenge to control ALB, our results not only demonstrate the response of a poplar species to wood borer infestation but also present an alternative method for poplar protection.

## 1. Introduction

Asian longhorned beetle *Anoplophora glabripennis* (ALB) adults emerge from June to August to supplement nutrition, mating, and oviposition in northern China. Adults gnaw notch grooves on host tree stems and oviposit one egg within each groove, and the hatched larvae eating the phloem and xylem severely destroy the physical structure and physiological function of the tree, which can cause the whole tree to die. Its host species are abundant and mainly include *Acer*, *Populus*, *Salix*, *Ulmus*, *Betula*, etc., [1,2,3]. The northwestern area in China is in arid and semi-arid region, where only a few tree species are able to be planted as a windbreak shelterbelt forest. *Populus alba* var. *pyramidalis* (PaP) is the main tree species found there because of its high height, narrow crown, salt and alkali resistance, and resistance to ALB to some extent. However, PaP can also be infested by ALB when there is no sensitive tree, so it is urgent to find effective methods to protect PaP [4,5].

Plants have evolved a complex defense system during the process of long-term plant and insect coevolution, including both constitutive and inducible defenses [6]. Upon insect infestation, the expression of a series of resistance genes is initiated through the plant’s hormone system, including the production of secondary defense substances and insect-resistant defense enzymes [7,8,9,10,11,12]. Jasmonic acid (JA) is a plant hormone that can be catalyzed by a series of enzymes to generate methyl jasmonate (MeJA). Mutual transformation of JA and MeJA results in signal transmission, transcription regulation, expression of related resistance genes, and synthesis of biochemical substances, thereby improving plant defense systems and resistance to infestation [13,14,15,16]. Salicylic acid (SA) is another important endogenous signal molecule responsible for a series of defense responses to diseases and pests. SA and SA-binding proteins can induce plant systems to exhibit system acquired resistance (SAR) that enhances innate plant resistance [17,18].

In recent years, studies have found that plants can form an immune memory called defense priming, and thus, exposure to trigger stimuli can respond faster or stronger to subsequent attacks of pests and pathogens. The initiation of this inducible defense provides increased protection and reduces the allocation cost of defense [19]. Application of plant hormones or other related substances like MeJA can induce or activate related signal transduction in plants, thereby improving the ability of plants to resist phytophagous insect damage [20,21,22]. We hypothesized that exogenous plant hormones could also trigger the defense system of PaP against ALB.

Volatiles emitted by host plants can mediate interspecific interactions between plants and insects [23]. Our previous results showed that (*Z*)-3-hexenol (Z3H) and (Z)-3-hexen-1-yl acetate (Z3HA) released from PaP were attractive to ALB [3,5]. We hypothesized that both volatiles might interact with the defense system of PaP, so in this study, we first analyzed the quantities of Z3H and Z3HA in the branches and leaves of healthy, mechanically damaged, and ALB-infested PaP. It is surprising that the quantities of the two volatiles from the latter two treatments were significantly lower than the healthy one, especially in the ALB-infested treatment. We investigated the synthesis pathways of Z3H and Z3HA through the Kyoto Encyclopedia of Genes and Genomes (KEGG) and found that both volatiles showed an antagonistic relationship with the synthesis of JA and SA. Then, we determined the contents of SA, JA, methyl salicylate (MeSA), and MeJA in the phloem of PaP either on healthy or ALB-infested conditions and found that both JA and MeJA were actually involved in the ALB-resistant response of PaP as a positive regulatory factor, especially MeJA. Finally, we verified the function of MeJA as a positive regulator of plant resistance to insect infestation by spraying exogenous MeJA on PaP seedlings, which could change some key defensive substances in PaP and affect the feeding and oviposition of ALB on PaP.

## 2. Materials and Methods

### 2.1. Collection and Identification of Volatiles Released by PaP

Volatiles were collected from the branches and leaves of 5-year-old PaP trees in situ (N: 40.341528; E: 106.927669). Trees in the field were divided into three groups, and each group had six trees. One group of trees was kept as a control. The trunk of each tree in the second group had ten notch grooves cut into it using a clean sterilized knife 1.2–1.7 m above the ground; this mechanical damage simulated ALB oviposition notch grooves. In the third group of trees, each trunk was caged with a nylon mesh (20 mesh) at 1.2–1.7 m above the ground, in which one adult male and three adult female ALB were introduced. After 10 notch grooves were gnawed on each tree by the females, the cages and the beetles were removed, and volatile collection began. Branches (30 cm length) were enclosed in a polyethylene (PE) bag (35 cm × 45 cm; Glad, ClCo., Ltd., Guangzhou, China), and a pumping system QC-1B (Beijing Labor Protection Scientific Research Institute, Beijing, China) was used to collect the released volatiles. Charcoal-filtered air at a flow rate of 450 mL/min was pumped into the PE bag, and the volatiles carried through a glass adsorption tube (10 cm length, 5 mm inner diameter) containing 200 mg of Porapak Q (80–100 mesh, Waters Corporation, Milford, MA, USA). Teflon tubes were used for all connections (Appendix A). Immediately after collection, volatiles were eluted from the trap tube with 1000 μL of dichloromethane and then concentrated down to 200 μL under a gentle flow of nitrogen at 0 °C and stored at −20 °C. Six replicate samples per treatment were used, each from a different tree. Empty bags were also considered as blank controls to enable possible contaminants to be identified and removed from analysis. Collections were made from 8:00 to 17:00 h, the active period of ALB adults.

Volatiles were identified using a GC capillary column (TG-WAX, 30 m × 0.25 mm × 0.25 µm) in an Agilent 7890B-7000C GC-MSD system (Agilent Technologies, Santa Clara, CA, USA). The injection volume of the samples was 1 μL. The injector port was at 240 °C, and the heating program was 40 °C for 1 min, an 8 °C/min increase to 180 °C, maintained for 1 min, a 20 °C/min increase to 200 °C, and maintained for 2 min. EI: 70 eV; MSD transmission line: 230 °C. The scanning mass range was 35–550 aum with a 0.2 s/scan. Primary identification was achieved by comparing the volatile compounds’ mass spectra with those from the NIST14 libraries. For bioactive compounds, commercial synthetic compounds were obtained to verify the retention time, the entire mass spectra, and the Kovats retention index, and the results were compared with those from the SDBS website at https://sdbs.db.aist.go.jp; (accessed on 15 July 2022). The quantities of both Z3H and Z3HA were calculated by standard solution curves already established in our previous studies [3].

### 2.2. KEGG Functional Annotation Analysis of Z3H and Z3HA Pathways

Z3H- and Z3HA-related gene pathway maps were queried with KEGG (https://www.genome.jp/kegg/; (accessed on 18 July 2022) to analyze upstream, downstream, and related regulatory genes (e.g., *LOX2S*, *AOS*, *HPL*). The biological information for related mediating and regulating genes were further obtained from NCBI (https://www.ncbi.nlm.nih.gov/; (accessed on 18 July 2022).

### 2.3. Identifying Insect-Resistant Endogenous Hormones in PaP

Ten individual healthy PaP trees of a similar diameter, height, and age were selected and divided into two groups. One group was infested with ALB, and the other was kept as a control. Infestation was achieved by caging ALB adults (five females and two males) on each trunk and allowing oviposition to proceed for 7 days, after which time eggs began hatching and 1st instar larvae began feeding on tree phloem. A total of 5 g of bark (3 cm × 3 cm) was sampled using a knife from 1.5 cm above an oviposition site on each infested tree, and 5 g of bark was taken from control group trees at a similar site on each tree trunk; this represented five replicates in each group. Before sampling, the sampling sites on all trees were washed with distilled water. Removed bark samples were immediately placed in liquid nitrogen and transported back to the laboratory on dry ice and stored at −80 °C for later testing.

Endogenous phytohormones were extracted from PaP samples using isopropanol–water–hydrochloric acid. The quantities of SA, JA, MeSA, and MeJA were determined on an Agilent 1290 HPLC (Agilent, USA) in series with an AB Sciex QTRAP 6500 + Mass Spectrometer (MS) (AB SCIEX, Framingham, MA, USA) and with an internal standard substance (2-hydroxybenzoic acid (2H-BA); dihydrojasmonic acid (2HJA); methyl dihydrojasmonate (2HMeJA) added to correct test results during the extraction process. Data were collected mainly using HPLC and MS/MS data acquisition systems.

Standard solutions of SA, JA, MeSA, and MeJA with gradients of 0.1, 0.2, 0.5, 2, 5, 20, 50, and 200 ng/mL were prepared using methanol (0.1% formic acid) as the solvent and included as internal standard solutions (2H-BA; 2HJA; 2HMeJA; 20 ng/mL).

The ESI-HPLC-MS/MS method was used for quantitative analysis of the plant hormones present. HPLC conditions were a chromatographic column, a Poroshell 120 SB-C18 reverse-phase chromatographic column (2.1 mm × 150 mm, 2.7 μm); a column temperature of 30 °C; reverse phase; and A:B = (methanol/0.1% formic acid):(water/0.1% formic acid). The elution gradient was 0–1 min, A = 20%; 1–9 min, A increased to 80%; 9–10 min, A = 80%; 10–10.1 min, and A decreased to 20%; 10.1–15 min, A = 20%, and the injection volume was 2 μL.

Tandem mass spectrometry (MS/MS) conditions were as follows: Electrospray Ionization Source (ESI) atomization temperature of 400 °C, curtain gas (CUR) at 15 psi, spray voltage (IS) at 4500 V, atomization gas pressure (Gas 1) at 65 psi, auxiliary gas pressure (Gas 2) at 70 psi, and the monitoring mode was MRM (multiple reaction monitoring mode). In the Q-Trap6500, each ion pair was scanned based on the optimized decluster voltage (DP) and collision energy (CE).

### 2.4. Activity of Defense Substances in the PaP Response to MeJA Treatment

Selected healthy (uninfested) PaP seedlings (two years old) were divided into three groups. Each group comprised fifteen trees. We first sprayed three replicate trees, each one treated with five concentrations (10^−3^, 10^−4^, 10^−5^, 10^−6^, 0 moL/L) of MeJA aqueous solutions [24,25]. After five days, quantities of H_2_O_2_, POD, and T-SOD were determined in 5 g of bark (3 cm × 3 cm) sampled from each tree in one group (group I) as a baseline control. Since taking samples was detrimental to the tree trunk, we did not sample this group again and regarded the other group (group III) as control. We then caged mated ALB adults on the trunks of PaP in the second group (group II) and allowed oviposition to proceed. Five adult females and two adult males per trunk were held together for 7 days previously to ensure they were mated and ready to oviposit. The third group (group III) of trees were maintained as controls with no ALB. Samples were taken 5 days after adult oviposition had been initiated in group II. A total of 5 g of bark (3 cm × 3 cm) was taken using a knife from 1.5 cm above the oviposition site in group II trees and, at the same time, 5 g of bark was taken from the group III controls at a similar position on the tree trunk. Since the determination and sampling were carried out twice, before and after pest infestation, two controls, I and III, were set. All samples were stored at −80 °C before further evaluation.

The quantities of H_2_O_2_, POD, and T-SOD in PaP bark in group I (different concentrations versus 0 mol/L MeJA aqueous solutions), group II (after MeJA treatment and ALB infestation), and group III (after MeJA treatment but no ALB infestation) were determined using a manufacturer’s kit (Nanjing Jiancheng Biotechnology Co., Ltd, Nanjing, China). The contents of H_2_O_2_, POD, and T-SOD were carried out according to the operation instructions of the reagent kit. Quantities were calculated according to the corresponding standard curve.

### 2.5. Collection and Identification of Volatiles from Leaves of PaP After MeJA Treatment

Different concentrations of MeJA (200 mL) (10^−3^, 10^−4^, 10^−5^, 10^−6^, 0 mol/L, five replicates per concentration) were sprayed per tested PaP tree (5 years old, different to the above tested trees) totally on leaves and trunks with household sprayers. Soil absorption was avoided. The spraying was carried out at 7:00 to 9:00 in the morning and on days without wind. Headspace solid-phase microextraction–gas chromatography–mass spectrometry (HS-SPME-GC-MS) was used to detect volatiles from branches and leaves. Fresh PaP (5-year-old) leaves (5 g) were collected from trees treated with different concentrations (10^−3^, 10^−4^, 10^−5^, 10^−6^, 0 mol/L, five replicates per concentration) of MeJA aqueous solutions, and each sample was placed in a clean 500 mL triangular conical flask. After 30 min of acclimation, an SPME with a DVB/CAR/PDMS (divinylbenzene/carboxyethyl/polydimethylsiloxane) extraction head (Supelco Co., Shanghai, China) was used to extract volatiles from each sample for 30 min at 25 °C. The chromatographic column was a DB-WAX device (30 m × 250 μm × 0.25 μm). The heating program was as follows: 40 °C for 1 min, 8 °C/min increase to 180 °C, maintained for 1 min, then 20 °C/min to 200 °C, and maintained for 2 min. Other GC-MS conditions were consistent with the identification conditions of host volatile components shown in Section 2.1.

### 2.6. Choices Made by ALB Adults for Branches and Leaves of PaP After Treatment with Different Concentrations of MeJA Aqueous Solutions

One-year-old healthy laboratory-cultivated PaP seedlings were each sprayed with a different concentration of MeJA aqueous solution (10^−3^, 10^−4^, 10^−5^, 10^−6^, 0 mol/L) at a rate of 200 mL per seedling; additional trees were mechanically damaged. Seedlings were labelled to indicate which treatment they had received. Two arenas (each with a 2 m diameter) were established, and one PaP seedling treated with each MeJA concentration and one mechanically injured PaP seedling were evenly arranged at six points around the circumference of each arena (Appendix A). Either ALB males (36 individuals) or ALB females (36 individuals) were released at the center of each arena. A steel screen cage (2 m × 2 m × 1.5 m) was placed over each arena to prevent the beetles from escaping. After 24 h, the number of beetles perched on each PaP seedling, the number of bites on branches and leaves, the feeding area, and the amount of feeding were recorded for each seedling. There were three replicate arenas for males and three for females, and the position of each treatment seedling in every test was rotated 120° amongst the three replicates to avoid directional bias in adults.

The damage level (*D*) of the ALB was calculated as follows: *D* = number of beetles selecting a particular treatment × 0.25 + number of bite wounds × 0.25 + food intake × 0.25 + amount of insect feces produced in 48 h × 0.25. The smaller the damage value was, the lower the damage level.

Food intake was calculated by weighing branches and leaves before and after ALB feeding. The amount of insect feces was calculated by weighing every beetle’s fresh feces produced in 24 h. The feces was held individually in plastic cages for 24 h after the arena tests were finished.

### 2.7. Oviposition of ALB Females on Trees Treated with Different Concentrations of MeJA Aqueous Solutions in the Field

Individual five-year-old healthy PaP trees (*n* = 25) were selected in the field, numbered, and divided into five groups. Trunks of replicate (*n* = 5) PaP trees in each group were each evenly sprayed with 200 mL of MeJA aqueous solutions (either 10^−3^, 10^−4^, 10^−5^, 10^−6^ or 0 mol/L) at 60–150 cm above the ground (Appendix A). After 24 h, the trunk of each tree was enclosed in a cage (10 mesh, 80–130 cm from the ground), and unmated ALB adults (one male and three females) were introduced into each cage. Fresh branches of PaP were placed into the cages daily to provide nutrition. Adult ALB were removed from the cages after 10 days. One month later, the oviposition sites on each trunk were excised and the number of notch grooves, surviving larvae, and weight of surviving larvae, were recorded.

Numbers of notch grooves, oviposition rate, and larval survival rate are important indices for evaluating offspring population density. Here, we propose a comprehensive evaluation index (*I*) to evaluate this value as follows: *I* = (number of notch grooves) × (oviposition rate) × (larval survival rate). The lower this index, the fewer offspring of ALB, and from the tree point, the smaller the index, the stronger was the host tree resistance. Larval weight was also recorded as a possible indicator of the offspring fitness. 

### 2.8. Statistics

The paired sample *t* test was used to compare the hormone content between ALB-infested and healthy PaP and the feeding results between males and females. One-way ANOVA (Tukey’s HSD) was used to compare differences in H_2_O_2_, POD, T-SOD, volatile content, and oviposition results, with treatments being considered significantly different at *p* < 0.05. The Origin 2018 software was used for mapping. Before analysis, raw data were tested for normality and homogeneity of variance. The SPSS 26.0 software was used for analysis.

## 3. Results

### 3.1. Comparisons of the Quantities of Z3HA and Z3H

The current study successfully collected and identified a total of 28 compounds (Figure 1; Appendix A). Since Z3HA and Z3H exhibited strong attractiveness towards ALB in a previous study, both chemicals were further studied in this paper.

Analysis showed that the quantity of Z3HA (*p* < 0.01) and Z3H (*p* < 0.01) in individual PaP trees following both mechanical injury and ALB infestation were lower than in healthy PaP individuals (Table 1). 

### 3.2. Relationships Between Z3HA, Z3H, and Synthesis Pathways of SA, JA, MeSA, and MeJA

The KEGG of the map 00592 pathway results indicated that Z3HA and Z3H were synthesized by a series of gene and enzyme reactions starting from α-linolenic acid released from cell membranes and that 13S-hydroperoxylinolenic acid was the key node of the Z3HA, Z3H and SA, JA, MeSA, and MeJA synthesis pathways (KEGG (https://www.genome.jp/kegg/; (accessed on 18 July 2022))(Figure 2; Appendix A). Thus, both pathways started from the same original metabolite (α-linolenic acid), which then divided into divergent routes. 

### 3.3. Quantity of SA, JA, MeSA, and MeJA in Healthy and ALB-Infested Plants

To determine whether plant hormones are involved in resistance of PaP to ALB, insect-resistance-related hormones were further quantified in healthy and ALB-infested trees (Table 2) (Appendix A). The results showed that the content of SA in PaP bark decreased significantly after ALB infestation (*t* = −40.13, *df* = 2, *p* < 0.01). The JA content increased after ALB infestation (*t* = 6.79, *df* = 2, *p* < 0.05). There was no significant difference in the quantity of MeSA before and after ALB infestation (*t* = −3.48, *df* = 2, *p* > 0.05), while the quantity of MeJA was significantly higher after ALB infestation (3.02 times greater than healthy trees, *t* = 10.06, *df* = 2, *p* < 0.05).

### 3.4. Activity of Defense Proteases After MeJA Treatment

To understand the role of MeJA in insect resistance by PaP, the quantities of H_2_O_2_, POD, and T-SOD in the bark of PaP were compared between healthy and ALB-infested PaP trees treated with concentrations of MeJA aqueous solutions. Since the quantities of H_2_O_2_, POD, and T-SOD in the two control groups (I and III; no ALB) were similar, we have not shown this comparison in Figure 3. As shown in Figure 3, at all concentrations of MeJA aqueous solutions, the H_2_O_2_ values in the ALB infestation (group II) treatment were significantly higher than in the healthy control group (group III) (*p* < 0.01). The POD content was also higher in the ALB-infested group (II) compared with the uninfested group (III); the POD content in the 10^−5^ mol/L treatment group increased by up to 2.59 times. The T-SOD quantities in the ALB-infested trees (group II) increased to 56.62% at 10^−6^ mol/L MeJA aqueous solution compared with the same concentration in uninfested trees (group III).

### 3.5. Quantity of Attractive Volatiles Released from PaP After MeJA Treatment

There were significant differences between the treatment groups in terms of the release of Z3HA (*p* < 0.01) and Z3H (*p* < 0.01), respectively. The lowest quantities of Z3HA and Z3H detected corresponded to the 10^−4^ mol/L MeJA aqueous solution treatment group (Table 3).

### 3.6. Feeding of ALB Adults on Branches and Leaves of PaP After MeJA Treatment Compared with Untreated Control and Mechanical Injury of PaP

The lowest food intake and least damage by ALB females were observed in the 10^−4^ mol/L MeJA aqueous solution treatment (Table 4). For male ALB, the 10^−3^ mol/L and 10^−4^ mol/L MeJA aqueous solution treatments led to the lowest food intake. The untreated and mechanical injury trees underwent similar damage by female adults, but male adults caused less damage on the mechanical injury trees in 48 h, almost the same level as the injury induced with the MeJA treatments. The feeding area and damage level of PaP branches and leaves were both lower in the MeJA treatments than in the control for both female and male ALB. The food intake of mechanical injury was between the untreated and MeJA treatment groups. The results indicate that the 10^−4^ mol/L treatment induced the strongest resistance in the host trees.

### 3.7. Analysis of ALB Oviposition on MeJA-Treated PaP

The results showed that the survival rate of ALB larvae on PaP treated with MeJA was significantly lower than that of untreated ones (*p* < 0.01). The comprehensive evaluation index (*I*) can effectively reflect ALB-resistance of PaP treated with MeJA aqueous solutions, with the concentration of 10^−4^ mol/L of MeJA inducing the highest ALB resistance (*p* < 0.01) (Table 5).

## 4. Discussion

Chemical inducers, including plant hormones, are a class of chemicals that make plants initiate defense responses to various stresses. These inducers can activate related signal transduction in plants or increase the sensitivity of triggering defense responses, thereby improving the ability of plants to resist herbivorous insect attacks [16,20,21]. MeJA is a plant hormone that has been used to artificially induce a variety of plants to refuse insect aggressions [26,27]. Our results confirmed that MeJA can effectively reduce damage in a poplar species by ALB.

Studies showed that plant volatile organic compounds (VOCs) and signal transduction networks in plants are closely related [28]. In our study, the quantities of SA and MeSA in PaP phloem significantly decreased after ALB infestation compared with controls, while the quantities of JA and MeJA in PaP phloem significantly increased (Table 2). In this context, Gong et al. (2023) [29] found that MeSA was involved in signaling pathways mediating volatile-driven defense against aphids and viruses. Because the quantities of Z3H and Z3HA released from PaP were significantly reduced after ALB infestation (Table 1) whilst JA and MeJA levels increased, we propose that ALB infestation activates the tree defense system to avoid further attack by subsequent attracted ALB adults.

MeJA can induce plants to produce secondary substances such as alkaloids and phenolic acids in response to insect feeding [30,31]. MeJA treatment of pepper is known to enhance resistance to *Myzus persicae,* increasing the activity of POD, SOD, and PPO; a 0.2 mM MeJA concentration was the best for avoiding invasion by *M. persicae* [9]. In this study, POD, T-SOD, and H_2_O_2_ activities also increased after MeJA treatment.

Since Farmer and Ryan (1990) reported that exogenous MeJA could induce plant insect resistance, inductance by JA and MeJA has received great attention, and a large amount of research has been carried out [25,32,33]. MeJA affected the emission of volatile terpenes by maritime pine (*Pinus pinaster* Aiton) and Monterey pine (*Pinus radiata* D. Don), which inhibited pine weevil feeding [24]. The growth and development of *Sitobion avenae* (Fabricius) and *Mythimna separata* feeding on MeJA-sprayed wheat seedlings hindered them and reduced their body weight [34,35]. MeJA treatment can also effectively induce insect resistance in *Cucumis sativus* and significantly reduced the survival of *Frankliniella occidentalis* [36]. MeJA can affect growth and detoxification mechanisms in *Spodoptera litura* [37]. Our results showed that the quantities of Z3H and Z3HA, which can attract ALB [3], decreased after MeJA treatment (Table 3), and the degree of feeding by ALB was also significantly reduced (Table 4). Spraying with MeJA aqueous solutions also reduced oviposition by ALB adults and larval fitness on PaP (Table 5), especially at a concentration of 10^−4^ mol/L.

Plants can develop an immune memory called defense priming, whereby exposure to trigger stimuli enables a faster or stronger response to subsequent attacks by pests and pathogens. Initiation of this induced defense provides increased protection and reduces the allocation cost of defense [19]. In this study, defense priming by MeJA was verified in a subsequent experiment investigating ALB selection and feeding on PaP as well as activating defense proteases in the tree. Compared with the control, the degree of damage on PaP treated with MeJA was reduced. It was dose-dependent, and different concentrations of MeJA aqueous solutions induced the emission of different quantities of Z3HA and Z3H, with the concentration of 10^−4^ mol/L MeJA aqueous solution inducing the greatest insect resistance in PaP. Therefore, JA and MeJA, especially MeJA, play a significant role in the defense response of PaP to ALB.

Exogenous MeJA can also enhance drought resistance in plants [35], which is particularly important for PaP in arid or semi-arid areas in China. However, how long the effects of MeJA persist requires more investigation. Moreover, application techniques for MeJA should also be studied, particularly the development of a sustainable release of MeJA at a particular rate, since plant responses are concentration-dependent.

## 5. Conclusions

Our results showed that after MeJA application, PaP had enhanced resistance to ALB. The resistance of PaP induced by the exogenous application of MeJA at a concentration of 10^−4^ mol/L was higher than at any other concentration tested.

## Figures and Tables

**Figure 1 insects-16-00153-f001:**
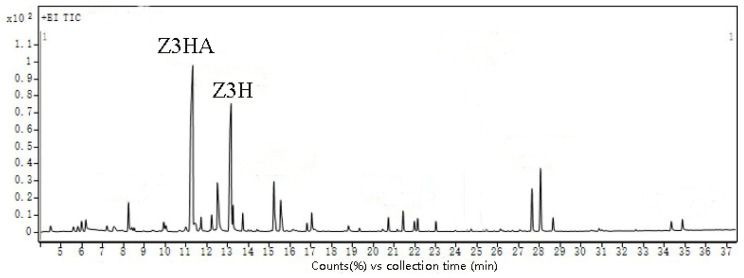
Total ion current chromatogram of volatile components of branches and leaves of *Populus alba* var. *pyramidalis*.

**Figure 2 insects-16-00153-f002:**
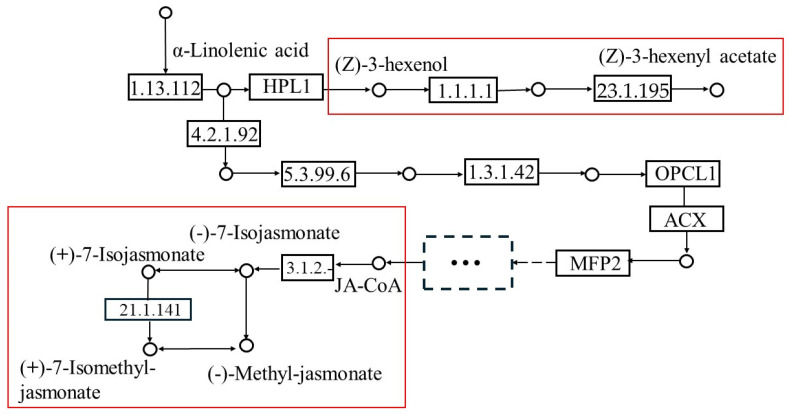
Z3HA, Z3H, and the metabolic pathways of SA, JA, MeSA, and MeJA. Pathways in the red boxes refer to our research targets.

**Figure 3 insects-16-00153-f003:**
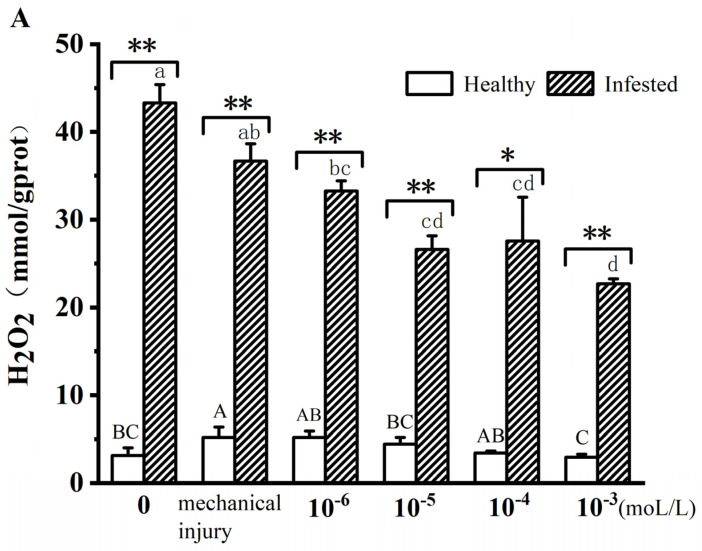
Comparison of three defense proteases in the bark of PaP between healthy and ALB-infested trees treated with different concentrations of MeJA aqueous solutions. (**A**) H_2_O_2_. (**B**) POD. (**C**) T-SOD. Note: Data are means ± SE. Healthy refers to group III and infested refers to group II. One-way ANOVA and Tukey’s HSD test were used to compare the difference between treatments with different letters (uppercase letters represent comparisons between healthy groups; lowercase letters represent comparisons between infested groups. *p* < 0.05). Paired sample *t* test compared the difference between healthy and infested trees for each treatment, ** *p* < 0.01; * *p* < 0.05; no marks mean no difference.

**Table 1 insects-16-00153-t001:** Quantities of Z3HA and Z3H released from PaP after mechanical injury and ALB infestation.

Volatiles	Concentration per Injection Volume (ng/µL)		
	Healthy (Control)	After Mechanical Injury	After ALB Infestation
Z3HA	1159.64 ± 3.27 a	577.82 ± 2.03 b	313.26 ± 3.11 c
Z3H	2049.12 ± 2.05 a	241.89 ± 2.18 c	521.81 ± 2.01 b

Note: Data are means ± SE (each group of trees). Data in the same row followed by different lowercase letters were significantly different at α = 0.05 following one-way ANOVA. PaP, *Populus alba* var. *pyramidalis.*

**Table 2 insects-16-00153-t002:** Amount of insect resistance hormones in the bark of healthy and infested PaP trees.

Phytohormone	Healthy or Infested by ALB	Quantity (ng/g)	Significance Level
SA	infested	17.79 ± 0.23	*p <* 0.01
	healthy	28.47 ± 0.16	
JA	infested	0.65 ± 0.09	*p <* 0.05
	healthy	0.31 ± 0.03	
MeSA	infested	2.13 ± 0.14	*p >* 0.05
	healthy	2.49 ± 0.03	
MeJA	infested	1.99 ± 0.13	*p* = 0.010
	healthy	0.66 ± 0.07	

Note: Data are means ± SE. A paired sample *t* test was used. *p* < 0.05 indicates difference was significant.

**Table 3 insects-16-00153-t003:** Comparison of quantities of Z3HA and Z3H released in different treatment groups.

Chemicals	Control	10^−3^ mol/L	10^−4^ mol/L	10^−5^ mol/L	10^−6^ mol/L
Z3HA	275.92 ± 1.43 a	210.54 ± 2.01 c	37.39 ± 1.50 e	218.17 ± 1.96 b	172.19 ± 2.01 d
Z3H	362.54 ± 2.17 b	34.11 ± 1.50 d	17.15 ± 1.37 e	400.49 ± 2.25 a	329.09 ± 10.57 c

Note: Data are means ± SE. Quantified by external standard method: Z3HA: Y = 1.3 X × 10^−6^ + 2.12; Z3H: Y = 3.72 X × 10^−6^ − 1.243; unit: ng/µL. Values in the same row followed by different lower-case letters were significantly different to each other (*p* < 0.05) (Tamhane’s multiple comparison).

**Table 4 insects-16-00153-t004:** Feeding of ALB adults on branches and leaves of PaP after MeJA treatment compared with untreated control and mechanical injury of PaP.

Gender	Treatment	Number of ALB Feeding	Feeding Area (mm^2^)	Food Intake (g)	Insect Frass in 48 h (mg)	Damage Level (*D*)
Female	Untreated	6.33 ± 1.53 a	9.00 ± 2.01 ab	0.52 ± 0.06 a	15.06 ± 0.41 a	7.73
	Mechanical injury	6.67 ± 2.89 a	10.33 ± 2.51 a	0.47 ± 0.18 a	11.40 ± 0.83 a	7.72
	10^−3^	7.33 ± 5.15 a	5.00 ± 1.00 bc	0.24 ± 0.08 ab	12.09 ± 3.98 a	6.17
	10^−4^	3.00 ± 0.00 a	3.67 ± 0.58 c	0.08 ± 0.01 b	7.56 ± 3.07 a	3.58
	10^−5^	3.67 ± 2.52 a	3.00 ± 1.00 c	0.09 ± 0.06 b	8.49 ± 1.42 a	3.81
	10^−6^	2.67 ± 0.58 a	3.33 ± 0.57 c	0.11 ± 0.01 b	11.11 ± 1.78 a	4.31
Male	Untreated	10.33 ± 1.20 a	16.67 ± 2.60 a	0.23 ± 0.04 a	4.72 ± 1.09 a	7.99
	Mechanical injury	4.00 ± 1.15 b	8.33 ± 1.33 a	0.12 ± 0.02 a	6.42 ± 0.30 a	4.72
	10^−3^	4.67 ± 1.85 b	7.67 ± 3.84 a	0.06 ± 0.02 b	4.44 ± 2.51 a	4.56
	10^−4^	1.33 ± 0.33 b	10.00 ± 2.52 a	0.06 ± 0.01 b	5.83 ± 2.68 a	4.31
	10^−5^	4.00 ± 1.53 b	8.33 ± 0.33 a	0.08 ± 0.05 ab	3.28 ± 0.96 a	3.95
	10^−6^	3.67 ± 1.45 b	5.67 ± 3.18 a	0.07 ± 0.03 b	4.24 ± 0.80 a	3.41

Note: Data are means ± SE. One-way ANOVA was used to compare differences between treatments. Values in the same column followed by different lowercase letters were significantly different to each other at α = 0.05 for females and males, respectively.

**Table 5 insects-16-00153-t005:** Oviposition of ALB females and larval survival rate on trunks of PaP treated with MeJA aqueous solutions.

Different Concentrations of MeJA (mol/L)	Notch Groove	Oviposition Rate (%)	Larval Survival Rate (%)	Larval Weight (mg)	Comprehensive Evaluation Index (*I*)
0	29.00 ± 4.00 a	71.67 ± 6.01 a	49.33 ± 5.81 a	43.00 ± 6.00 a	10.25
10^−3^	11.33 ± 1.76 bc	72.22 ± 10.23 a	27.21 ± 2.26 bc	24.00 ± 5.00 c	2.23
10^−4^	9.33 ± 1.33 bc	68.73 ± 3.61 a	22.07 ± 4.16 c	33.00 ± 3.00 b	1.42
10^−5^	6.67 ± 2.18 c	66.21 ± 4.52 a	37.22 ± 3.89 b	36.00 ± 3.00 b	1.63
10^−6^	16.33 ± 3.66 b	73.20 ± 10.69 a	21.16 ± 1.96 c	44.00 ± 5.00 a	2.53

Note: Data are means ± SE. One-way ANOVA was used to compare differences between treatments. Values in the same column followed by different lowercase letters were significantly different to each other at α = 0.05.

## Data Availability

The data presented in this study are available on request from the corresponding author.

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
