# Peer review of "Methyl Jasmonate Enhances the Resistance of Populus alba var. pyramidalis Against Anoplophora glabripennis (Coleoptera: Cerambycidae)"

_insects, 2025, doi:10.3390/insects16020153_

Round 1
Reviewer 1 Report
Comments and Suggestions for Authors
The manuscript by Shao et al. investigates the resistance of Populus alba var. pyramidalis (PaP) to Anoplophora glabripennis (ALB) infestation using exogenous treatment with a defense-inducing compound. The research found that ALB infestation reduced certain volatile compounds in PaP and altered plant hormone levels, with significant increases in jasmonic acid (JA) and methyl jasmonate (MeJA), along with elevated defense substances like H2O2 and POD. Exogenous spraying of MeJA reduced ALB feeding and oviposition on PaP, suggesting that MeJA enhances resistance. The findings highlight MeJA’s potential as a strategy for improving PaP resistance to ALB. These findings are promising and useful for pest management. The study is well-organized, detailed, and complete. I have some minor comments and suggestions for the authors to consider.
Major Comments:
The abstract can be better organized by emphasizing some of the methodological details of the studies performed throughout this project. The manuscript contains several sub-studies, and their details are not clear from the abstract.
Miscellaneous suggestions:
L66: "Defence initiation" or "defence priming"? If I am not mistaken, the correct term here should be "priming."
L103: I assume you used a pumping system as a headspace collector. The "push-pull" system has a different meaning; please correct this. Also, provide the trade name of the pump or manufacturer, etc.
Figures 1 and 2: These could be moved to supplementary files. Please use entomological terminology, such as "gravid female," instead of "pregnant female."
L272: Figure 4 is not directly linked to the core study and shows the widely known metabolic pathways of MeSA, SA, JA, and MeJA. Therefore, this figure could be better placed as a supplementary file or deleted.
Figure 5: I suggest the authors consult with a biostatistician to verify the statistical analysis methods used for these treatments.
L317: I recommend providing the degrees of freedom in the widely accepted format, as "df = 4, 10," rather than using "(df1 and df2)." This will ensure clarity and consistency in presenting the statistical results.
L322: Please provide the statistical test used for this comparison here.
L327-328: Please rewrite this sentence as: "The lowest food intake and least damage by ALB females were observed in the 10⁻⁴ mol/L MeJA aqueous solution treatment," for clarity. Also, avoid using "smallest," and prefer "lowest" throughout the manuscript.
L354-358: Rather than presenting Gong et al. (2023) findings first, please relate this information to your own findings in the discussion.
L374: Please check all Latin names and italicize them.
More studies on the exogenous application of methyl jasmonates under field conditions have been conducted. I suggest the authors cite additional earlier papers as a foundation for their findings, as well as recommend the use of MeJA under field conditions for controlling various pests.
Author Response
- The abstract can be better organized by emphasizing some of the methodological details of the studies performed throughout this project. The manuscript contains several sub-studies, and their details are not clear from the abstract. …………………
Response: Thanks for your suggestion. We have added some methodological details, please check the revised abstract.
- L66: "Defence initiation" or "defence priming"? If I am not mistaken, the correct term here should be "priming." …………………
Response: Thanks for your comment. We revised this error.
- L103: I assume you used a pumping system as a headspace collector. The "push-pull" system has a different meaning; please correct this. Also, provide the trade name of the pump or manufacturer, etc. Figures 1 and 2: These could be moved to supplementary files. Please use entomological terminology, such as "gravid female," instead of "pregnant female." ……………………………………
Response: Thanks for your suggestions. W added the manufacture name and revised the terminology, please see the revised ms. We deleted both fig.1 and fig 2 in the main text of manuscript, and moved them into a supplement file, please check the supplement file for both figures.
- L272: Figure 4 is not directly linked to the core study and shows the widely known metabolic pathways of MeSA, SA, JA, and MeJA. Therefore, this figure could be better placed as a supplementary file or deleted. ……………………………………
Response: Great thanks for your suggestion. We moved it into the supplement file.
- Figure 5: I suggest the authors consult with a biostatistician to verify the statistical analysis methods used for these treatments. ………………………………………………………
Response: Thanks for your suggestions. We checked again for the original data and analysis method, and asked help from statistician and revised some wrong description in the 2.8 statistics part.
- L317: I recommend providing the degrees of freedom in the widely accepted format, as "df = 4, 10," rather than using "(df1 and df2)." This will ensure clarity and consistency in presenting the statistical results. ………………………………………………………
Response: Thanks for your suggestion, and we corrected them in the revised ms.
- L322: Please provide the statistical test used for this comparison here. ……………………
Response: Thanks, and we added the analysis methods here, please check the revision edition.
- L327-328: Please rewrite this sentence as: "The lowest food intake and least damage by ALB females were observed in the 10⁻⁴ mol/L MeJA aqueous solution treatment," for clarity. Also, avoid using "smallest," and prefer "lowest" throughout the manuscript. …………………
Response: Thanks, we adopted your writing and please see the revised ms.
- L354-358: Rather than presenting Gong et al. (2023) findings first, please relate this information to your own findings in the discussion. ……………………………………
Response: Thanks, and we delated this redundant sentence.
- L374: Please check all Latin names and italicize them. …………………
Response: Thanks, and we corrected it.
- More studies on the exogenous application of methyl jasmonates under field conditions have been conducted. I suggest the authors cite additional earlier papers as a foundation for their findings, as well as recommend the use of MeJA under field conditions for controlling various pests. ……………………………………
Response: Thanks, and we added some new references.
Reviewer 2 Report
Comments and Suggestions for Authors
Although this manuscript had been well written, it still has some points to be improved. It had been illustrated as follows:
Introduction section:
1. It's better for adding the contents of exact responses of defense phytohormones signaling pathways, such as direct treatment could alter the HIPV of plants or trigger the diverse responses.
2. In my opinion, the last paragraph should be improved, it is not the discussion.
Methods: It's better for you guys to describe the reasons why you selected these treatment concentrations.
Results:
1. The Figure and table contents need the descripation in detail.
2. Providing the high-resolution figure 1 and removing the unavailable information.
3. Preparing the new figure 4 with clear and simple pathways, instead of current figure 4. It's better for you placing the original figure into the supplementary files.
4. Providing the statistic results on 3.3 section.
Author Response
Introduction section:
- It's better for adding the contents of exact responses of defense phytohormones signaling pathways, such as direct treatment could alter the HIPV of plants or trigger the diverse responses. ………………………………………………………
Response: Thank you for your suggestions. We did think adding more detailed information in the introduction part, and cited the references from 20-22, and 24, 25, pleased see the revised ms. However, since there are many published papers dealing with this issue and it is a well-known knowledge in plant indirect defense but seldom in the tree system, so we condensed them in the third paragraph of the introduction part and did not exaggerate it more. Thanks for your kindly consideration.
- In my opinion, the last paragraph should be improved, it is not the discussion. …………………
Response: Thanks for your suggestion. We rewrote this paragraph a bit and make it clearer as the objective of the ms.
Methods:
It's better for you guys to describe the reasons why you selected these treatment concentrations. …………………
Response: Thank you for your comments and we added the references here. Please checked the third line in the 2.4 part.
Results:
1.The figure and table contents need the descripation in detail.……………………………
Response: Thank you for your suggestions and we checked and adding more information in the figures and tables.
- Providing the high-resolution figure 1 and removing the unavailable information. …………
Response: Thank you for this comment. According to the first reviewer, we moved the original figure 1 to the supplement file.
- Preparing the new figure 4 with clear and simple pathways, instead of current figure 4. It's better for you placing the original figure into the supplementary files. …………………
Response: Thank you for your suggestion. According to the suggestion of both reviewers, we moved it forward to the supplement file.
- Providing the statistic results on 3.3 section. …………………
Response: Thanks for your suggestions and we added the P values. The original table 6 is move forward to supplement file.
Round 2
Reviewer 2 Report
Comments and Suggestions for Authors
Results: The Figure and table contents need the descripition in detail.
1. 3.1 and 3.2 missing the the statistic results in detail.
2. Providing the high-resolution figure 1 and removing the unavailable information.
3. Preparing the new figure 4 with clear and simple pathways for 3.2.
4. Providing the statistic results in detail on 3.3 section.
5. So did in 3.7.
Author Response
Dear Editor and Reviewers,
Great thanks for your comments and suggestion for our manuscript again. Appended to this letter is our point-by-point response to the comments raised by you. The comments are reproduced and our responses are given directly below.
We would like also to thank you for allowing us to revise the manuscript.
Best Regards,
Jian-Rong Wei, Peng-Peng Shao
Response to Comments:
Results:
1.“3.1 and 3.2 missing the statistic results in detail..” ………………
Response: Thanks for your suggestion, and we added them accordingly.
2.“Providing the high-resolution figure 1 and removing the unavailable information.” ………
Response: Thanks for your advice, and we revised the figure 1.
3.Preparing the new figure 4 with clear and simple pathways for 3.2." ………………………Response: Thanks for your suggestions. We revised this figure and put it into the supplement file. Please see the sup-figure 3. In the first revision edition, according the reviewer advice, we move this figure into the supplement file. In the revised figure, we deleted some unnecessary routes.
- Providing the statistic results in detail on 3.3 section.. ……………………………………
Response: Great thanks for your suggestion. We added the statistics results.
- So did in 3.7. ………………………………………………………
Response: Thanks for your suggestions, and we added the statistics results.
Round 3
Reviewer 2 Report
Comments and Suggestions for Authors
The authors had been improved the manuscript.
Author Response
Thank you very much for your review.